# Recurrently Controlled Recurrent Networks

**Yi Tay[1], Luu Anh Tuan[2], and Siu Cheung Hui[3]**
[1,3]Nanyang Technological University
[2]Institute for Infocomm Research
ytay017@e.ntu.edu.sg[1]
at.luu@i2r.a-star.edu.sg[2]
asschui@ntu.edu.sg[3]

## Abstract

Recurrent neural networks (RNNs) such as long short-term memory and gated recurrent units are pivotal building blocks across a broad spectrum of sequence modeling problems. This paper proposes a recurrently controlled recurrent network (RCRN) for expressive and powerful sequence encoding. More concretely, the key idea behind our approach is to learn the recurrent gating functions using recurrent networks. Our architecture is split into two components - a controller cell and a listener cell whereby the recurrent controller actively influences the compositionality of the listener cell. We conduct extensive experiments on a myriad of tasks in the NLP domain such as sentiment analysis (SST, IMDb, Amazon reviews, etc.), question classification (TREC), entailment classification (SNLI, SciTail), answer selection (WikiQA, TrecQA) and reading comprehension (NarrativeQA). Across all 26 datasets, our results demonstrate that RCRN not only consistently outperforms BiLSTMs but also stacked BiLSTMs, suggesting that our controller architecture might be a suitable replacement for the widely adopted stacked architecture.

## 1 Introduction

Recurrent neural networks (RNNs) live at the heart of many sequence modeling problems. In particular, the incorporation of gated additive recurrent connections is extremely powerful, leading to the pervasive adoption of models such as Gated Recurrent Units (GRU) [Cho et al., 2014] or Long Short-Term Memory (LSTM) [Hochreiter and Schmidhuber, 1997] across many NLP applications [Bahdanau et al., 2014; Xiong et al., 2016; Rocktäschel et al., 2015; McCann et al., 2017]. In these models, the key idea is that the gating functions control information flow and compositionality over time, deciding how much information to read/write across time steps. This not only serves as a protection against vanishing/exploding gradients but also enables greater relative ease in modeling long-range dependencies.

There are two common ways to increase the representation capability of RNNs. Firstly, the number of hidden dimensions could be increased. Secondly, recurrent layers could be stacked on top of each other in a hierarchical fashion [El Hihi and Bengio, 1996], with each layer's input being the output of the previous, enabling hierarchical features to be captured. Notably, the wide adoption of stacked architectures across many applications [Graves et al., 2013; Sutskever et al., 2014; Wang et al., 2017; Nie and Bansal, 2017] signify the need for designing complex and expressive encoders. Unfortunately, these strategies may face limitations. For example, the former might run a risk of overfitting and/or hitting a wall in performance. On the other hand, the latter might be faced with the inherent difficulties of going deep such as vanishing gradients or difficulty in feature propagation across deep RNN layers [Zhang et al., 2016b].

This paper proposes Recurrently Controlled Recurrent Networks (RCRN), a new recurrent architecture and a general purpose neural building block for sequence modeling. RCRNs are characterized by

its usage of two key components - a recurrent controller cell and a listener cell. The *controller* cell controls the information flow and compositionality of the *listener* RNN. The key motivation behind RCRN is to provide expressive and powerful sequence encoding. However, unlike stacked architectures, all RNN layers operate jointly on the same hierarchical level, effectively avoiding the need to go deeper. Therefore, RCRNs provide a new alternate way of utilizing multiple RNN layers in conjunction by allowing one RNN to control another RNN. As such, our key aim in this work is to show that our proposed controller-listener architecture is a viable replacement for the widely adopted stacked recurrent architecture.

To demonstrate the effectiveness of our proposed RCRN model, we conduct extensive experiments on a plethora of diverse NLP tasks where sequence encoders such as LSTMs/GRUs are highly essential. These tasks include sentiment analysis (SST, IMDb, Amazon Reviews), question classification (TREC), entailment classification (SNLI, SciTail), answer selection (WikiQA, TrecQA) and reading comprehension (NarrativeQA). Experimental results show that RCRN outperforms BiLSTMs and multi-layered/stacked BiLSTMs on all **26** datasets, suggesting that RCRNs are viable replacements for the widely adopted stacked recurrent architectures. Additionally, RCRN achieves close to state-of-the-art performance on several datasets.

## 2 Related Work

RNN variants such as LSTMs and GRUs are ubiquitous and indispensible building blocks in many NLP applications such as question answering [Seo et al., 2016; Wang et al., 2017], machine translation [Bahdanau et al., 2014], entailment classification [Chen et al., 2017] and sentiment analysis [Longpre et al., 2016; Huang et al., 2017]. In recent years, many RNN variants have been proposed, ranging from multi-scale models [Koutnik et al., 2014; Chung et al., 2016; Chang et al., 2017] to tree-structured encoders [Tai et al., 2015; Choi et al., 2017]. Models that are targetted at improving the internals of the RNN cell have also been proposed [Xingjian et al., 2015; Danihelka et al., 2016]. Given the importance of sequence encoding in NLP, the design of effective RNN units for this purpose remains an active area of research.

Stacking RNN layers is the most common way to improve representation power. This has been used in many highly performant models ranging from speech recognition [Graves et al., 2013] to machine reading [Wang et al., 2017]. The BCN model [McCann et al., 2017] similarly uses multiple BiLSTM layers within their architecture. Models that use shortcut/residual connections in conjunctin with stacked RNN layers are also notable [Zhang et al., 2016b; Longpre et al., 2016; Nie and Bansal, 2017; Ding et al., 2018].

Notably, a recent emerging trend is to model sequences without recurrence. This is primarily motivated by the fact that recurrence is an inherent prohibitor of parallelism. To this end, many works have explored the possibility of using attention as a replacement for recurrence. In particular, self-attention [Vaswani et al., 2017] has been a popular choice. This has sparked many innovations, including general purpose encoders such as DiSAN [Shen et al., 2017] and Block Bi-DiSAN [Shen et al., 2018]. The key idea in these works is to use multi-headed self-attention and positional encodings to model temporal information.

While attention-only models may come close in performance, some domains may still require the complex and expressive recurrent encoders. Moreover, we note that in [Shen et al., 2017, 2018], the scores on multiple benchmarks (e.g., SST, TREC, SNLI, MultiNLI) do not outperform (or even approach) the state-of-the-art, most of which are models that still heavily rely on bidirectional LSTMs [Zhou et al., 2016; Choi et al., 2017; McCann et al., 2017; Nie and Bansal, 2017]. While self-attentive RNN-less encoders have recently been popular, our work moves in an orthogonal and possibly complementary direction, advocating a stronger RNN unit for sequence encoding instead. Nevertheless, it is also good to note that our RCRN model outperforms DiSAN in all our experiments.

Another line of work is also concerned with eliminating recurrence. SRUs (Simple Recurrent Units) [Lei and Zhang, 2017] are recently proposed networks that remove the sequential dependencies in RNNs. SRUs can be considered a special case of Quasi-RNNs [Bradbury et al., 2016], which performs incremental pooling using pre-learned convolutional gates. A recent work, Multi-range Reasoning Units (MRU) [Tay et al., 2018b] follows the same paradigm, trading convolutional gates with features learned via expressive multi-granular reasoning. Zhang et al. [2018] proposed sentence-state LSTMs (S-LSTM) that exchanges incremental reading for a single global state.

Our work proposes a new way of enhancing the representation capability of RNNs without going deep. For the first time, we propose a controller-listener architecture that uses one recurrent unit to control another recurrent unit. Our proposed RCRN consistently outperforms stacked BiLSTMs and achieves state-of-the-art results on several datasets. We outperform above-mentioned competitors such as DiSAN, SRUs, stacked BiLSTMs and sentence-state LSTMs.

## 3  Recurrently Controlled Recurrent Networks (RCRN)

This section formally introduces the RCRN architecture. Our model is split into two main components - a controller cell and a listener cell. Figure 1 illustrates the model architecture.

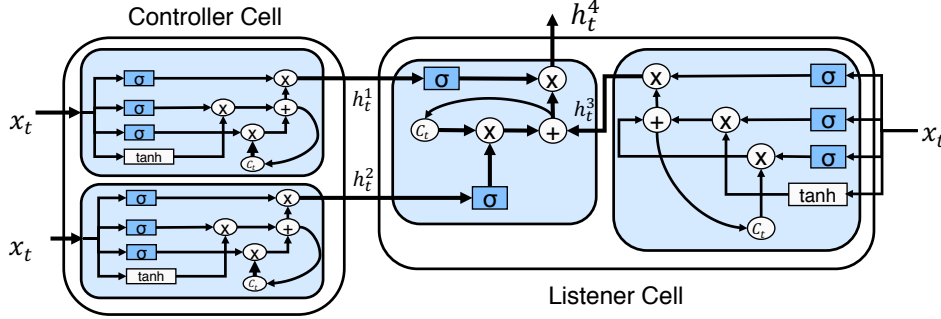

Figure 1: High level overview of our proposed RCRN architecture.

### 3.1  Controller Cell

The goal of the controller cell is to learn gating functions in order to influence the target cell. In order to control the target cell, the controller cell constructs a forget gate and an output gate which are then used to influence the information flow of the listener cell. For each gate (output and forget), we use a separate RNN cell. As such, the controller cell comprises two cell states and an additional set of parameters. The equations of the controller cell are defined as follows:

$$i_t^1 = \sigma_s(W_i^1 x_t + U_i^1 h_{t-1}^1 + b_i^1) \text{ and } i_t^2 = \sigma_s(W_i^2 x_t + U_i^2 h_{t-1}^2 + b_i^2) \tag{1}$$

$$f_t^1 = \sigma_s(W_f^1 x_t + U_f^1 h_{t-1}^1 + b_f^1) \text{ and } f_t^2 = \sigma_s(W_f^2 x_t + U_f^2 h_{t-1}^2 + b_f^2) \tag{2}$$

$$o_t^1 = \sigma_s(W_o^1 x_t + U_o^1 h_{t-1}^1 + b_o^1) \text{ and } o_t^2 = \sigma_s(W_o^2 x_t + U_o^2 h_{t-1}^2 + b_o^2) \tag{3}$$

$$c_t^1 = f_t^1 \, c_{t-1}^1 + i_t^1 \, \sigma(W_c^1 x_t + U_c^1 h_{t-1}^1 + b_c^1) \tag{4}$$

$$c_t^2 = f_t^2 \, c_{t-1}^2 + i_t^2 \, \sigma(W_c^2 x_t + U_c^2 h_{t-1}^2 + b_c^2) \tag{5}$$

$$h_t^1 = o_t^1 \odot \sigma(c_t^1) \text{ and } h_t^2 = o_t^2 \odot \sigma(c_t^2) \tag{6}$$

where $x_t$ is the input to the model at time step $t$. $W_*^k, U_*^k, b_*^k$ are the parameters of the model where $k = \{1, 2\}$ and $* = \{i, f, o\}$. $\sigma_s$ is the sigmoid function and $\sigma$ is the tanh nonlinearity. $\odot$ is the Hadamard product. The controller RNN has two cell states denoted as $c^1$ and $c^2$ respectively. $h_t^1, h_t^2$ are the outputs of the unidirectional controller cell at time step $t$. Next, we consider a bidirectional adaptation of the controller cell. Let Equations (1-6) be represented by the function CT(), the bidirectional adaptation is represented as:

$$\overrightarrow{h_t^1}, \overrightarrow{h_t^2} = \overrightarrow{\text{CT}}(h_{t-1}^1, h_{t-1}^2, x_t) \quad t = 1, \cdots \ell \tag{7}$$

$$\overleftarrow{h_t^1}, \overleftarrow{h_t^2} = \overleftarrow{\text{CT}}(h_{t+1}^1, h_{t+1}^2, x_t) \quad t = M, \cdots 1 \tag{8}$$

$$h_t^1 = [\overrightarrow{h_t^1}; \overleftarrow{h_t^1}] \text{ and } h_t^2 = [\overrightarrow{h_t^2}; \overleftarrow{h_t^2}] \tag{9}$$

The outputs of the bidirectional controller cell are $h_t^1, h_t^2$ for time step $t$. These hidden outputs act as gates for the listener cell.

## 3.2 Listener Cell

The listener cell is another recurrent cell. The final output of the RCRN is generated by the listener cell which is being influenced by the controller cell. First, the listener cell uses a base recurrent model to process the sequence input. The equations of this base recurrent model are defined as follows:

$$i_t^3 = \sigma_s(W_i^3 x_t + U_i^3 h_{t-1}^3 + b_i^3) \tag{10}$$

$$f_t^3 = \sigma_s(W_f^3 x_t + U_f^3 h_{t-1}^3 + b_f^3) \tag{11}$$

$$o_t^3 = \sigma_s(W_o^3 x_t + U_o^3 h_{t-1}^3 + b_o^3) \tag{12}$$

$$c_t^3 = f_t^3 \, c_{t-1}^3 + i_t^3 \, \sigma(W_c^3 x_t + U_c^3 h_{t-1}^3 + b_c^3) \tag{13}$$

$$\overrightarrow{h_t^3} = o_t^3 \odot \sigma(c_t^3) \tag{14}$$

Similarly, a bidirectional adaptation is used, obtaining $h_t^3 = [\overrightarrow{h_t^3}, \overleftarrow{h_t^3}]$. Next, using $h_t^1, h_t^2$ (outputs of the controller cell), we define another recurrent operation as follows:

$$c_t^4 = \sigma_s(h_t^1) \odot c_{t-1}^4 + (1 - \sigma_s(h_t^1)) \odot h_t^3 \tag{15}$$

$$h_t^4 = h_t^2 \odot c_t^3 \tag{16}$$

where $c_t^j, h_t^j$ and $j = \{3, 4\}$ are the cell and hidden states at time step $t$. $W_*^3, U_*^3$ are the parameters of the listener cell where $* = \{i, f, o\}$. Note that $h_t^1$ and $h_t^2$ are the outputs of the controller cell. In this formulation, $\sigma_s(h_t^1)$ acts as the forget gate for the listener cell. Likewise $\sigma_s(h_t^2)$ acts as the output gate for the listener.

## 3.3 Overall RCRN Architecture, Variants and Implementation

Intuitively, the overall architecture of the RCRN model can be explained as follows: Firstly, the controller cell can be thought of as two BiRNN models which hidden states are used as the forget and output gates for another recurrent model, i.e., the listener. The listener uses a single BiRNN model for sequence encoding and then allows this representation to be altered by listening to the controller. An alternative interpretation to our model architecture is that it is essentially a 'recurrent-over-recurrent' model. Clearly, the formulation we have used above uses BiLSTMs as the atomic building block for RCRN. Hence, we note that it is also possible to have a simplified variant[1] of RCRN that uses GRUs as the atomic block which we found to have performed slightly better on certain datasets.

**Cuda-level Optimization**  For efficiency purposes, we use the CUDNN optimized version of the base recurrent unit (LSTMs/GRUs). Additionally, note that the final recurrent cell (Equation (15)) can be subject to CUDA-level optimization[2] following simple recurrent units (SRU) [Lei and Zhang, 2017]. The key idea is that this operation can be performed along the dimension axis, enabling greater parallelization on the GPU. For the sake of brevity, we refer interested readers to [Lei and Zhang, 2017]. Note that this form of cuda-level optimization was also performed in the Quasi-RNN model [Bradbury et al., 2016], which effectively subsumes the SRU model.

**On Parameter Cost and Memory Efficency**  Note that a single RCRN model is equivalent to a stacked BiLSTM of 3 layers. This is clear when we consider how two controller BiRNNs are used to control a single listener BiRNN. As such, for our experiments, when considering only the encoder and keeping all other components constant, 3L-BiLSTM has equal parameters to RCRN while RCRN and 3L-BiLSTM are approximately three times larger than BiLSTM.

## 4 Experiments

This section discusses the overall empirical evaluation of our proposed RCRN model.

## 4.1 Tasks and Datasets

In order to verify the effectiveness of our proposed RCRN architecture, we conduct extensive experiments across several tasks[3] in the NLP domain.

**Sentiment Analysis**   Sentiment analysis is a text classification problem in which the goal is to determine the polarity of a given sentence/document. We conduct experiments on both sentence and document level. More concretely, we use 16 Amazon review datasets from [Liu et al., 2017], the well-established Stanford Sentiment TreeBank (SST-5/SST-2) [Socher et al., 2013] and the IMDb Sentiment dataset [Maas et al., 2011]. All tasks are binary classification tasks with the exception of SST-5. The metric is the accuracy score.

**Question Classification**   The goal of this task is to classify questions into fine-grained categories such as *number* or *location*. We use the TREC question classification dataset [Voorhees et al., 1999]. The metric is the accuracy score.

**Entailment Classification**   This is a well-established and popular task in the field of natural language understanding and inference. Given two sentences $s_1$ and $s_2$, the goal is to determine if $s_2$ entails or contradicts $s_1$. We use two popular benchmark datasets, i.e., the Stanford Natural Language Inference (SNLI) corpus [Bowman et al., 2015], and SciTail (Science Entailment) [Khot et al., 2018] datasets. This is a pairwise classssification problem in which the metric is also the accuracy score.

**Answer Selection**   This is a standard problem in information retrieval and learning-to-rank. Given a question, the task at hand is to rank candidate answers. We use the popular WikiQA [Yang et al., 2015] and TrecQA [Wang et al., 2007] datasets. For TrecQA, we use the cleaned setting as denoted by Rao et al. [2016]. The evaluation metrics are the MAP (Mean Average Precision) and Mean Reciprocal Rank (MRR) ranking metrics.

**Reading Comprehension**   This task involves reading documents and answering questions about these documents. We use the recent NarrativeQA [Kočiskỳ et al., 2017] dataset which involves reasoning and answering questions over story summaries. We follow the original paper and report scores on BLEU-1, BLEU-4, Meteor and Rouge-L.

## 4.2 Task-Specific Model Architectures and Implementation Details

In this section, we describe the task-specific model architectures for each task.

**Classification Model**   This architecture is used for all text classification tasks (sentiment analysis and question classification datasets). We use 300D GloVe [Pennington et al., 2014] vectors with 600D CoVe [McCann et al., 2017] vectors as pretrained embedding vectors. An optional character-level word representation is also added (constructed with a standard BiGRU model). The output of the embedding layer is passed into the RCRN model directly without using any projection layer. Word embeddings are not updated during training. Given the hidden output states of the $200d$ dimensional RCRN cell, we take the concatenation of the max, mean and min pooling of all hidden states to form the final feature vector. This feature vector is passed into a single dense layer with ReLU activations of $200d$ dimensions. The output of this layer is then passed into a softmax layer for classification. This model optimizes the cross entropy loss. We train this model using Adam [Kingma and Ba, 2014] and learning rate is tuned amongst $\{0.001, 0.0003, 0.0004\}$.

**Entailment Model**   This architecture is used for entailment tasks. This is a pairwise classification models with two input sequences. Similar to the singleton classssification model, we utilize the identical input encoder (GloVe, CoVE and character RNN) but include an additional part-of-speech (POS tag) embedding. We pass the input representation into a two layer highway network [Srivastava et al., 2015] of 300 hidden dimensions before passing into the RCRN encoder. The feature representation of $s1$ and $s2$ is the concatenation of the max and mean pooling of the RCRN hidden outputs. To compare $s1$ and $s2$, we pass $[s_1, s_2, s_1 \odot s_2, s_1 - s_2]$ into a two layer highway network. This output

is then passed into a softmax layer for classification. We train this model using Adam and learning rate is tuned amongst $\{0.001, 0.0003, 0.0004\}$. We mainly focus on the *encoder-only* setting which does not allow cross sentence attention. This is a commonly tested setting on the SNLI dataset.

**Ranking Model**   This architecture is used for the ranking tasks (i.e., answer selection). We use the model architecture from Attentive Pooling BiLSTMs (AP-BiLSTM) [dos Santos et al., 2016] as our base and swap the RNN encoder with our RCRN encoder. The dimensionality is set to $200$. The similarity scoring function is the cosine similarity and the objective function is the pairwise hinge loss with a margin of $0.1$. We use negative sampling of $n = 6$ to train our model. We train our model using Adadelta [Zeiler, 2012] with a learning rate of $0.2$.

**Reading Comprehension Model**   We use R-NET [Wang et al., 2017] as the base model. Since R-NET uses three Bidirectional GRU layers as the encoder, we replaced this stacked BiGRU layer with RCRN. For fairness, we use the GRU variant of RCRN instead. The dimensionality of the encoder is set to $75$. We train both models using Adam with a learning rate of $0.001$.

For all datasets, we include an additional ablative baselines, swapping the RCRN with (1) a standard BiLSTM model and (2) a stacked BiLSTM of 3 layers (3L-BiLSTM). This is to fairly observe the impact of different encoder models based on the same overall model framework.

### 4.3   Overall Results

This section discusses the overall results of our experiments.

| Dataset/Model | BiLSTM | 2L-BiLSTM | SLSTM | BiLSTM$^\dagger$ | 3L-BiLSTM$^\dagger$ | RCRN |
|---|---|---|---|---|---|---|
| Camera | 87.1 | 88.1 | 90.0 | 87.3 | 89.7 | **90.5** |
| Video | 84.7 | 85.2 | 86.8 | 87.5 | 87.8 | **88.5** |
| Health | 85.5 | 85.9 | 86.5 | 85.5 | 89.0 | **90.5** |
| Music | 78.7 | 80.5 | 82.0 | 83.5 | 85.7 | **86.0** |
| Kitchen | 82.2 | 83.8 | 84.5 | 81.7 | 84.5 | **86.0** |
| DVD | 83.7 | 84.8 | 85.5 | 84.0 | 86.0 | **86.8** |
| Toys | 85.7 | 85.8 | 85.3 | 87.5 | 90.5 | **90.8** |
| Baby | 84.5 | 85.5 | 86.3 | 85.0 | 88.5 | **89.0** |
| Books | 82.1 | 82.8 | 83.4 | 86.0 | 87.2 | **88.0** |
| IMDB | 86.0 | 86.6 | 87.2 | 86.5 | 88.0 | **89.8** |
| MR | 75.7 | 76.0 | 76.2 | 77.7 | 77.7 | **79.0** |
| Apparel | 86.1 | 86.4 | 85.8 | 88.0 | 89.2 | **90.5** |
| Magazines | 92.6 | 92.9 | 93.8 | 93.7 | 92.5 | **94.8** |
| Electronics | 82.5 | 82.3 | 83.3 | 83.5 | 87.0 | **89.0** |
| Sports | 84.0 | 84.8 | 85.8 | 85.5 | 86.5 | **88.0** |
| Software | 86.7 | 87.0 | 87.8 | 88.5 | 90.3 | **90.8** |
| Macro Avg | 84.3 | 84.9 | 85.6 | 85.7 | 87.5 | **88.6** |

Table 1: Results on the Amazon Reviews dataset. $^\dagger$ are models implemented by us.

| Model/Reference | Acc |
|---|---|
| MVN [Guo et al., 2017] | 51.5 |
| DiSAN [Shen et al., 2017] | 51.7 |
| DMN [Kumar et al., 2016] | 52.1 |
| LSTM-CNN [Zhou et al., 2016] | 52.4 |
| NTI [Yu and Munkhdalai, 2017] | 53.1 |
| BCN [McCann et al., 2017] | 53.7 |
| BCN + ELMo [Peters et al., 2018] | **54.7** |
| BiLSTM | 51.3 |
| 3L-BiLSTM | 52.6 |
| RCRN | 54.3 |

Table 2: Results on Sentiment Analysis on SST-5.

| Model/Reference | Acc |
|---|---|
| P-LSTM [Wieting et al., 2015] | 89.2 |
| CT-LSTM [Looks et al., 2017] | 89.4 |
| TE-LSTM [Huang et al., 2017] | 89.6 |
| NSE [Munkhdalai and Yu, 2016] | 89.7 |
| BCN [McCann et al., 2017] | 90.3 |
| BMLSTM [Radford et al., 2017] | **91.8** |
| BiLSTM | 89.7 |
| 3L-BiLSTM | 90.0 |
| RCRN | 90.6 |

Table 3: Results on Sentiment Analysis on SST-2.

**Sentiment Analysis**   On the 16 review datasets (Table 1) from [Liu et al., 2017; Zhang et al., 2018], our proposed RCRN architecture achieves the highest score on all 16 datasets, outperforming the existing state-of-the-art model - sentence state LSTMs (SLSTM) [Zhang et al., 2018]. The

| Model/Reference | Acc |
|---|---|
| Res. BiLSTM [Longpre et al., 2016] | 90.1 |
| 4L-QRNN [Bradbury et al., 2016] | 91.4 |
| BCN [McCann et al., 2017] | 91.8 |
| oh-LSTM [Johnson and Zhang, 2016] | 91.9 |
| TRNN [Dieng et al., 2016] | 93.8 |
| Virtual Miyato et al. [2016] | **94.1** |
| BiLSTM | 90.9 |
| 3L-BiLSTM | 91.8 |
| RCRN | 92.8 |

Table 4: Results on IMDb binary sentiment clasification.

| Model/Reference | Acc |
|---|---|
| CNN-MC [Kim, 2014] | 92.2 |
| SRU [Lei and Zhang, 2017] | 93.9 |
| DSCNN [Zhang et al., 2016a] | 95.4 |
| DC-BiLSTM [Ding et al., 2018] | 95.6 |
| BCN [McCann et al., 2017] | 95.8 |
| LSTM-CNN [Zhou et al., 2016] | 96.1 |
| BiLSTM | 95.8 |
| 3L BiLSTM | 95.4 |
| RCRN | **96.2** |

Table 5: Results on TREC question classification.

| Model/Reference | Acc |
|---|---|
| Multi-head [Vaswani et al., 2017] | 84.2 |
| Att. Bi-SRU [Lei and Zhang, 2017] | 84.8 |
| DiSAN [Shen et al., 2017] | 85.6 |
| Shortcut [Nie and Bansal, 2017] | 85.7 |
| Gumbel LSTM [Choi et al., 2017] | 86.0 |
| Dynamic Meta Emb [Kiela et al., 2018] | **86.7** |
| BiLSTM | 85.5 |
| 3L-BiLSTM | 85.1 |
| RCRN | 85.8 |

Table 6: Results on SNLI dataset.

| Model/Reference | Acc |
|---|---|
| ESIM [Chen et al., 2017] | 70.6 |
| DecompAtt [Parikh et al., 2016] | 72.3 |
| DGEM [Khot et al., 2018] | 77.3 |
| CAFE [Tay et al., 2017] | 83.3 |
| CSRAN [Tay et al., 2018a] | 86.7 |
| OpenAI GPT [Radford et al., 2018] | **88.3** |
| BiLSTM | 80.1 |
| 3L-BiLSTM | 79.6 |
| RCRN | 81.1 |

Table 7: Results on SciTail dataset.

| | WikiQA | | TrecQA | |
|---|---|---|---|---|
| Model | MAP | MRR | MAP | MRR |
| BiLSTM | 68.5 | 69.8 | 72.4 | 82.5 |
| 3L-BiLSTM | 69.3 | 71.3 | 73.0 | 83.6 |
| RCRN | 71.1 | 72.3 | 75.4 | 85.5 |
| AP-BiLSTM | 63.9 | 69.9 | 75.1 | 80.0 |
| AP-3L-BiLSTM | 69.8 | 71.3 | 73.3 | 83.4 |
| AP-RCRN | **72.4** | **73.7** | **77.9** | **88.2** |

Table 8: Results on Answer Retrieval (WikiQA and TrecQA).

| Model | Bleu-1 | Bleu-4 | Meteor | Rouge |
|---|---|---|---|---|
| Seq2Seq | 16.1 | 1.40 | 4.2 | 13.3 |
| ASR | 23.5 | 5.90 | 8.0 | 23.3 |
| BiDAF | 33.7 | 15.5 | 15.4 | 36.3 |
| R-NET | 34.9 | 20.3 | 18.0 | 36.7 |
| RCRN | **38.1** | **21.8** | **18.1** | **38.3** |

Table 9: Results on Reading Comprehension (NarrativeQA).

macro average performance gain over BiLSTMs ($+4\%$) and Stacked (2 X BiLSTM) ($+3.4\%$) is also notable. On the same architecture, our RCRN outperforms ablative baselines BiLSTM by $+2.9\%$ and 3L-BiLSTM by $+1.1\%$ on average across 16 datasets.

Results on SST-5 (Table 2) and SST-2 (Table 3) are also promising. More concretely, our RCRN architecture achieves state-of-the-art results on SST-5 and SST-2. RCRN also outperforms many strong baselines such as DiSAN [Shen et al., 2017], a self-attentive model and Bi-Attentive classification network (BCN) [McCann et al., 2017] that also use CoVe vectors. On SST-2, strong baselines such as Neural Semantic Encoders [Munkhdalai and Yu, 2016] and similarly the BCN model are also outperformed by our RCRN model.

Finally, on the IMDb sentiment classification dataset (Table 4), RCRN achieved $92.8\%$ accuracy. Our proposed RCRN outperforms Residual BiLSTMs [Longpre et al., 2016], 4-layered Quasi Recurrent Neural Networks (QRNN) [Bradbury et al., 2016] and the BCN model which can be considered to be very competitive baselines. RCRN also outperforms ablative baselines BiLSTM ($+1.9\%$) and 3L-BiLSTM ($+1\%$).

**Question Classification**    Our results on the TREC question classification dataset (Table 5) is also promising. RCRN achieved a state-of-the-art score of $96.2\%$ on this dataset. A notable baseline is the Densely Connected BiLSTM [Ding et al., 2018], a deep residual stacked BiLSTM model which RCRN outperforms ($+0.6\%$). Our model also outperforms BCN (+0.4%) and SRU (+2.3%). Our ablative BiLSTM baselines achieve reasonably high score, posssibly due to CoVe Embeddings. However, our RCRN can further increase the performance score.

**Entailment Classification**    Results on entailment classification are also optimistic. On SNLI (Table 6), RCRN achieves 85.8% accuracy, which is competitive to Gumbel LSTM. However, RCRN outperforms a wide range of baselines, including self-attention based models as multi-head [Vaswani et al., 2017] and DiSAN [Shen et al., 2017]. There is also performance gain of +1% over Bi-SRU even though our model does not use attention at all. RCRN also outperforms shortcut stacked encoders, which use a series of BiLSTM connected by shortcut layers. Post review, as per reviewer request, we experimented with adding cross sentence attention, in particular adding the attention of Parikh et al. [2016] on 3L-BiLSTM and RCRN. We found that they performed comparably (both at $\approx 87.0$). We did not have resources to experiment further even though intuitively incorporating different/newer variants of attention [Kim et al., 2018; Tay et al., 2018a; Chen et al., 2017] and/or ELMo [Peters et al., 2018] can definitely raise the score further. However, we hypothesize that cross sentence attention forces less reliance on the encoder. Therefore stacked BiLSTMs and RCRNs perform similarly.

The results on SciTail similarly show that RCRN is more effective than BiLSTM (+1%). Moreover, RCRN outperforms several baselines in [Khot et al., 2018] including models that use cross sentence attention such as DecompAtt [Parikh et al., 2016] and ESIM [Chen et al., 2017]. However, it still falls short to recent state-of-the-art models such as OpenAI's Generative Pretrained Transformer [Radford et al., 2018].

**Answer Selection**    Results on the answer selection (Table 8) task show that RCRN leads to considerable improvements on both WikiQA and TrecQA datasets. We investigate two settings. The first, we reimplement AP-BiLSTM and swap the BiLSTM for RCRN encoders. Secondly, we completely remove all attention layers from both models to test the ability of the standalone encoder. Without attention, RCRN gives an improvement of $+ \approx 2\%$ on both datasets. With attentive pooling, RCRN maintains a $+ \approx 2\%$ improvement in terms of MAP score. However, the gains on MRR are greater $(+4 - 7\%)$. Notably, AP-RCRN model outperforms the official results reported in [dos Santos et al., 2016]. Overall, we observe that RCRN is much stronger than BiLSTMs and 3L-BiLSTMs on this task.

**Reading Comprehension**    Results (Table 9) show that enhancing R-NET with RCRN can lead to considerable improvements. This leads to an improvement of $\approx 1\% - 2\%$ on all four metrics. Note that our model only uses a single layered RCRN while R-NET uses 3 layered BiGRUs. This empirical evidence might suggest that RCRN is a better way to utilize multiple recurrent layers.

**Overall Results**    Across **all** 26 datasets, RCRN outperforms not only standard BiLSTMs but also 3L-BiLSTMs which have approximately equal parameterization. 3L-BiLSTMs were overall better than BiLSTMs but lose out on a minority of datasets. RCRN outperforms a wide range of competitive baselines such as DiSAN, Bi-SRUs, BCN and LSTM-CNN, etc. We achieve (close to) state-of-the-art performance on SST, TREC question classification and 16 Amazon review datasets.

## 4.4   Runtime Analysis

This section aims to get a benchmark on model performance with respect to model efficiency. In order to do that, we benchmark RCRN along with BiLSTMs and 3 layered BiLSTMs (with and without CUDNN optimization) on different sequence lengths (i.e., $16, 32, 64, 128, 256$). We use the IMDb sentiment task. We use the same standard hardware (a single Nvidia GTX1070 card) and an identical overarching model architecture. The dimensionality of the model is set to 200 with a fixed batch size of 32. Finally, we also benchmark a CUDA optimized adaptation of RCRN which has been described earlier (Section 3.3).

Table 10 reports training/inference times of all benchmarked models. The fastest model is naturally the 1 layer BiLSTM (CUDNN). Intuitively, the speed of RCRN should be roughly equivalent to using 3 BiLSTMs. Surprisingly, we found that the CUDA optimized RCRN performs consistently slightly faster than the 3 layer BiLSTM (CUDNN). At the very least, RCRN provides comparable efficiency to using stacked BiLSTM and empirically we show that there is nothing to lose in this aspect. However, we note that CUDA-level optimizations have to be performed. Finally, the non-CUDNN optimized BiLSTM and stacked BiLSTMs are also provided for reference.

| | Training Time (seconds/epoch) | | | | | Inference (seconds/epoch) | | | | |
|---|---|---|---|---|---|---|---|---|---|---|
| | 16 | 32 | 64 | 128 | 256 | 16 | 32 | 64 | 128 | 256 |
| 3 layer BiLSTM | 29 | 50 | 113 | 244 | 503 | 12 | 20 | 38 | 72 | 150 |
| BiLSTM | 18 | 30 | 63 | 131 | 272 | 9 | 15 | 28 | 52 | 104 |
| 1 layer BiLSTM (CUDNN) | 5 | 6 | 9 | 14 | 26 | 2 | 3 | 4 | 6 | 10 |
| 3 layer BiLSTM (CUDNN) | 10 | 14 | 23 | 42 | 80 | 4 | 5 | 9 | 16 | 32 |
| RCRN (CUDNN) | 19 | 29 | 53 | 101 | 219 | 8 | 12 | 23 | 41 | 78 |
| RCRN (CUDNN +CUDA optimized) | 10 | 13 | 21 | 40 | 78 | 4 | 5 | 8 | 15 | 29 |

Table 10: Training and Inference times on IMDb binary sentiment classification task with varying sequence lengths.

## 5    Conclusion and Future Directions

We proposed Recurrently Controlled Recurrent Networks (RCRN), a new recurrent architecture and encoder for a myriad of NLP tasks. RCRN operates in a novel controller-listener architecture which uses RNNs to learn the gating functions of another RNN. We apply RCRN to a potpourri of NLP tasks and achieve promising/highly competitive results on all tasks and 26 benchmark datasets. Overall findings suggest that our controller-listener architecture is more effective than stacking RNN layers. Moreover, RCRN remains equally (or slightly more) efficient compared to stacked RNNs of approximately equal parameterization. There are several potential interesting directions for further investigating RCRNs. Firstly, investigating RCRNs controlling other RCRNs and secondly, investigating RCRNs in other domains where recurrent models are also prevalent for sequence modeling. The source code of our model can be found at `https://github.com/vanzytay/NIPS2018_RCRN`.

## 6    Acknowledgements

We thank the anonymous reviewers and area chair from NeurIPS 2018 for their constructive and high quality feedback.

## Footnotes

[1]We omit technical descriptions due to the lack of space.

[2]We adapt the CUDA kernel as a custom Tensorflow op in our experiments. While the authors of SRU release their cuda-op at `https://github.com/taolei87/sru`, we use a third-party open-source Tensorflow version which can be found at `https://github.com/JonathanRaiman/tensorflow_qrnn.git`.

[3]While we agree that other tasks such as language modeling or NMT would be interesting to investigate, we could not muster enough GPU resources to conduct any extra experiments. We leave this for future work.

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
