[Reviews · NeurIPS 2018]

Reviewer 1



Update: Thanks for your response. I'm glad to hear that you are going to open source the optimizations; I look forward to playing with this. Best of luck with the follow-up work, and I look forward to seeing how the RCRN performs on SNLI in the cross-attention task setting (really hoping to see this in the camera-ready!). Original Review: The core idea behind this paper is to use RNNs (LSTMs or GRUs in this work) to compute the gates (input, forget, etc.) for a higher level RNN. They use this idea to show how to improve performance on a large number of tasks (mostly classification) with relatively simple models that have little to no loss in efficiency compared to models that perform similarly. Some of the performances achieved new state-of-the-art results. When implemented well, they can even be faster despite the apparent addition of complexity. The setup as controller and listener cells seems unnecessarily confusing. It seems clearer to describe this as a collection of three RNNs, two that compute gates for the state produced by the third. If it was presented that way, then it would be clearer that Eq. 15 and 16 are really the meat of the RCRN. The first 14 equations are really just normal (or bidirectional adaptations of normal) RNN equations (7-9 are nice to see for clarity). I understand that it can be helpful to write out all of those equations some times, but I would recommend defining an RNN cell once, and then focus on describing how you use the three different states, rather than duplicating the same equations 2 times in Eq. 1-6 and then again in 10-14. On my first read, I found myself looking for potential difference in all those equations because I assumed they were written out so explicitly for that reason. In equations 1-3, not sure if it is a typo, but either way: it would be clearer if the W_i, W_f, and W_o used to get i_t^2, f_t^2, and o_t^2 respectively had a numerical superscript like the other matrices. It sounds like it would be a 2, but without the superscript it kind of feels like a typo since all the other matrices have superscripts. U_i^1 is also used for i_t^2, but it seems like that should be U_i^2 instead. Would like a little clarification or a note if these aren’t typos clarifying that some of the weights are indeed shared. Why not run on SQuAD? That is a pretty standard QA dataset now, so much so that its absence makes me wonder whether that was a failure case for RCRNs. It would be great if you could assuage that concern for future readers. I could almost say the same for MultiNLI over SNLI now as well, but I understand that both of these have external testing systems that can make using them difficult some times. Why do you only compare with models that do not use semi-supervised learning on external corpora? If you are using CoVe, which is using supervised learning on external corpora and GloVe, then it might seem a bit arbitrary to rule out semi-supervised augmentations in those tables. I think the claims are still effective for RCRN because the BiLSTM and 3l-BiLSTM also use CoVe, POS embeddings, etc. What exactly does CUDA optimized mean? How hard would these be to make on your own? Or will you be open sourcing these essential optimizations or even better building into something like PyTorch or Tensorflow? If you want maximum impact, definitely help people get that super fast CUDA optimized version. If that existed open source, I would definitely start trying it out. I’m not convinced about the claim that this is different than ‘going deeper’. This just seems like a different (apparently smarter/better) way of going deeper than by just stacking RNNs as layers. I wonder whether you could just use a single RNN cell as the Controller and do a simpler transformation of that state to get the different gate values. Perhaps this could be even faster? Can we have RCRNs controlling RCRNs? Character level embeddings are using a BiGRU model; is this better than using pertained character embeddings? Lots of the true SOTA models for entailment do use sentence cross attention, and you chose not to. Why? How do the RCRN’s purported benefits stand up when attention is in the mix for entailment? Or perhaps the Highway networks are particularly important and you don’t get any benefit from the RCRN without them? It seems insane at first glance that your 3L-BiLSTM is as good as all those other more complex models even without the RCRN. Is that coming from the GRU character embeddings or the POS embeddings or the CoVe embeddings? Embeddings are just really that important it seems, and as we’ve created better representations, simple BILSTMs and stacked versions are getting better.
 Overall is a 7, with a confidence of 4. I'm reserving very confident for reproducibility of systems that have been declared as being open source in the future. This submission includes a lot of detail, but it seems unlikely that everyone who wants to use this work will implement those custom CUDA ops themselves. The RCRN seems simple enough to reproduce, but at the speeds presented, it seems potentially difficult.

Reviewer 2



This submission proposes a new recurrent unit, namely, recurrently controlled recurrent networks (RCRN). It is inspired by LSTM, but the sigmoid gates and content cells are replaced with additional recurrent networks. In this way, the gates would not depend on the previous hidden states, using the notations from the paper, h^4, but are implemented as separate gating recurrent units instead. The method appears novel and interesting to me, and one can imagine there are straightforward extensions under such a framework, just as the extensive efforts the community have been devoted to improving gating RNNs. The paper conducts extensive empirical evaluations on multiple NLP tasks, and the proposed achieves strong performance. Last but not least, the paper is well-written and easy to follow. Details: - Although the experimental part is already quite extensive, I would be excited to see results on language modeling and machine translation. - The 3-layer LSTM might not be the most comparable baseline, since there is no vertical connection in a RCNRN. - The notations in the equations and those in Figure 1 are not consistent. - Some of the experimental details are missing.

Reviewer 3



The proposes to extend gating mechanisms with recurrency. The idea is extremely simple and, to be honest, I am shocked that it hasn’t been done before. The technical exposition is clear, which I attribute mostly to the simplicity of the architecture. It’s difficult to imagine any criticism of this paper: a simple idea with strong empirical results. In fact, the only argument I can see against accepting this paper is that it’s a bit boring. I wouldn’t be surprised if you couldn’t find this very idea in the footnotes of a handful of other papers. Questions: Do the authors actually control for parameters? The experiments demonstrate that the RCN network is better. Does it also have more parameters? I didn’t see this explicitly mentioned and it would indicate that the experiments aren’t fair. Why not use a task with strong auto-regressive tendencies? I would have loved to see a language-modeling experiment or a neural MT experiment. It’s hard to imagine any of the proposed tasks have strong interesting gating tendencies since bag-of-words baselines are very strong on simple classification tasks.